# Speech Separation based on pre-trained model and Deep Modularization

## Abstract

This paper introduces a novel speech separation technique that leverages the clustering of time-frequency (T-F) bin patches or raw speech blocks. Our approach integrates traditional graph-based clustering objectives with deep neural networks, enabling effective and scalable speech separation. By extracting features from T-F bin patches or raw speech blocks using a pre-trained encoder, we apply deep modularization for clustering, allowing us to identify clusters dominated by individual speakers in mixed speech signals. Extensive evaluations across multiple datasets, such as WSJ0-2mix and WHAM!, demonstrate the competitiveness of our method compared to fully supervised state-of-the-art speech separation models. In particular, our approach excels in separating complex acoustic mixtures without the need for parallel datasets and effectively mitigates the problem of permutation ambiguity, making it well-suited for real-world applications in multi-speaker environments.

## 1 Introduction

Speech separation is a challenging task, where the goal is to separate individual sources from a mixed speech signal. Many tools, including those proposed by (Zeghidour & Grangier, 2021a), (Weng et al., 2015), and (Hershey et al., 2016), frame this task as a multi-class regression problem. In this approach, training a speech separation model involves comparing its output to the target sources, with the model producing an output dimension for each target class. However, when multiple sources of the same type are present, the system must arbitrarily assign output dimensions to each target, leading to the well-known permutation ambiguity problem (Hershey et al., 2016). This ambiguity arises when the system cannot differentiate between identical sources, such as two overlapping speakers, and randomly assigns each source to an output dimension.

To address permutation ambiguity, various techniques have been developed, with the most notable being permutation invariant training (PIT), introduced by (Yu et al., 2017). PIT determines the optimal output-target pairing by minimizing the error between the predicted and target signals, regardless of the order in which they appear. While PIT has shown effectiveness, it suffers from high computational complexity, with factorial time complexity $O(S!)$, making it expensive as the number of sources ($S$) increases (Tachibana, 2021; Dovrat et al., 2021).

Furthermore, PIT-based models often encounter a mismatch between the number of speakers during training and inference due to their fixed output dimensions (Jiang & Duan, 2020). To mitigate this, these models set a maximum number of sources $S$ that they can output for any given mixture (Nachmani et al., 2020; Liu & Wang, 2019a). When a mixture contains fewer than $S$ sources ($S > K$), the model generates $S - K$ invalid outputs. Some models handle this by outputting silence for the invalid sources (Liu & Wang, 2019a; Nachmani et al., 2020), while others discard invalid outputs based on energy levels relative to the mixture (Luo & Mesgarani, 2020). However, these energy-based methods can struggle in low-energy scenarios, where accurately discarding invalid outputs becomes more difficult (Luo & Mesgarani, 2020). Another class of models (Shi et al., 2018; Takahashi et al., 2019) iteratively generates a single speech signal and subtracts it from the mixture until no speech remains. However, determining when to stop the iterations is non-trivial, and performance often degrades during later iterations (Takahashi et al., 2019).

Given the computational complexity and limitations of PIT-based models, clustering-based methods have emerged as a promising alternative for handling permutation ambiguity. Methods proposed by (Hershey et al., 2016), (Qin et al., 2020), and (Zeghidour & Grangier, 2021b) apply clustering techniques to separate multiple speakers from mixed speech signals. These approaches avoid the need for output-target pairing by grouping time-frequency (T-F) bins based on their similarity, assigning each bin to the speaker that dominates the given time-frequency region. However, despite their success, clustering-based methods still rely on large amounts of annotated speech data, which is costly and labor-intensive to produce.

In this work, we introduce a novel speech separation technique based on clustering, known as deep modularization (Müller, 2023), which eliminates the need for parallel annotated datasets. Deep modularization is a technique that combines traditional graph-based clustering objectives with deep neural networks to optimize feature partitioning in an end-to-end differentiable manner. We extract features from spectrogram points (T-F bins) or raw speech blocks using a pre-trained model, and leverage these features in downstream clustering tasks via deep modularization. This approach enables us to achieve speech separation without the need for parallel speech data, addressing one of the major bottlenecks in current speech separation systems.

Our contributions are as follows:

1. We propose a competitive clustering technique for speech separation that optimizes the cluster assignments of spectrogram points or raw speech blocks in an end-to-end differentiable manner, without relying on parallel data.

2. We evaluate the performance of the proposed technique using both raw speech and Fourier-transformed speech as input.

3. We conduct extensive evaluations across multiple datasets to assess the effectiveness of the proposed technique in various speech separation scenarios.

4. We perform ablation studies to investigate whether training the pre-trained model with contaminated inputs benefits the downstream task of speech separation.

## 1.1 Related Work

Several unsupervised speech separation methods have been developed in recent years. One notable technique is MixIT (Wisdom et al., 2020b), which performs speech separation by mixing mixtures. Given a set of mixed speech signals $X = x_1, x_2, \cdots, x_n$, where each mixture $x_i$ contains up to $N$ sources, MixIT creates a mixture of mixtures (MoM) by randomly drawing and adding mixtures from $X$. For instance, if $x_1$ and $x_2$ are selected, the MoM $\bar{x}$ is created as $\bar{x} = x_1 + x_2$. This MoM is then fed into a deep neural network (DNN), which estimates the sources $\hat{s}$ within $x_1$ and $x_2$. The model is trained to minimize the following loss:

$$L_{MixIT} = \min_A \sum_{i=1}^{2} L(x_i, [A\hat{s}]_i)$$

Here, $A \in B^{2 \times M}$ is a set of binary matrices, where each column sums to 1, and $M$ is the maximum number of sources in a given mixture $x_i$. The loss minimizes the difference between mixtures $x_i$ and the remixed separated sources $A\hat{s}$. A limitation of MixIT is over-separation, where the model estimates more sources than are present in the mixtures (Zhang et al., 2021). Additionally, MixIT struggles with denoising (Saito et al., 2021). Variants of MixIT, such as (Zhang et al., 2021) and (Karamatlı & Kırbız, 2022), address the over-separation problem, while (Trinh & Braun, 2022) and (Saito et al., 2021) enhance MixIT for denoising tasks.

Another unsupervised method, RemixIT (Tzinis et al., 2022), employs a teacher-student model for speech denoising and separation. The teacher model estimates the clean speech sources $\hat{s}_i$ and noises $\hat{n}_i$ for a batch of size $b$. The estimated noises are randomly mixed to generate $n^p$, which, together with the clean sources, form new mixtures $\hat{m}_i = \hat{s}_i + n^p$. These new mixtures are used as input to the student model, which is trained

to output $\hat{s}_i$ and $n^p$, effectively denoising the speech. RemixIT relies on a pre-trained speech enhancement model as the teacher.

Other unsupervised approaches, such as (Wang et al., 2016), perform speech separation based on speaker characteristics like gender. These methods use i-vectors to model differences in vocal tract properties, pitch, timing, and rhythm between male and female speakers, and the DNN acts as a gender separator.

Speech separation tools are designed to accept either Fourier spectrum or time-domain input features. Fourier spectrum-based models apply the discrete Fourier transform (DFT) to convert the raw time-domain signal into the frequency domain. These models recognize the non-stationary nature of speech and its variability in both time and frequency. Commonly used features include log-power spectrum (Fu et al., 2017; Xu et al., 2015), Mel-frequency spectrum (Liu et al., 2022; Du et al., 2020; Donahue et al., 2018), DFT magnitude (Nossier et al., 2020; Grais & Plumbley, 2018; Fu et al., 2019), and complex DFT features (Fu et al., 2017; Williamson & Wang, 2017; Kothapally & Hansen, 2022a;b). Most models using Fourier spectrum features assume that phase information is less critical for human auditory perception and primarily focus on magnitude or power for training (Xu et al., 2014; Kumar & Florencio, 2016; Du & Huo, 2008; Tu et al., 2014; Li et al., 2017). The noisy phase is typically used for signal reconstruction based on the work of (Ephraim & Malah, 1984), which suggests that the phase of the noisy signal is a close estimator of the clean signal's phase.

However, recent research (Paliwal et al., 2011) shows that further improvements in speech quality can be achieved by processing both the magnitude and phase spectra. To address phase challenges with Fourier-based features, several time-domain speech separation models have been developed, such as (Luo & Mesgarani, 2018; Luo et al., 2020; Luo & Mesgarani, 2019a; Subakan et al., 2021a). These models replace the DFT-based input with data-driven representations learned during model training. Instead of converting the raw signal to the frequency domain, they process the mixed waveform directly, generating either the estimated clean sources or masks applied to the noisy waveform to produce clean speech. This approach allows the models to fully capture both magnitude and phase information, leading to improved separation performance (Luo et al., 2020).

## 2  Speech Separation by Deep Modularization

In this section, we describe our proposed speech separation technique based on deep modularization. The process consists of four key components: 1) Speech pre-processing, where spectrogram points (T-F bins) or raw speech blocks are extracted from the input speech; 2) A pre-trained model that learns representations of these spectrogram points or raw blocks; 3) Deep modularization, which clusters the extracted features into $k$ clusters corresponding to different speakers; and 4) Clean signal generation, where $k$ clean speech signals are reconstructed from the clustered data. An overview of the entire pipeline is shown in Figure 1.

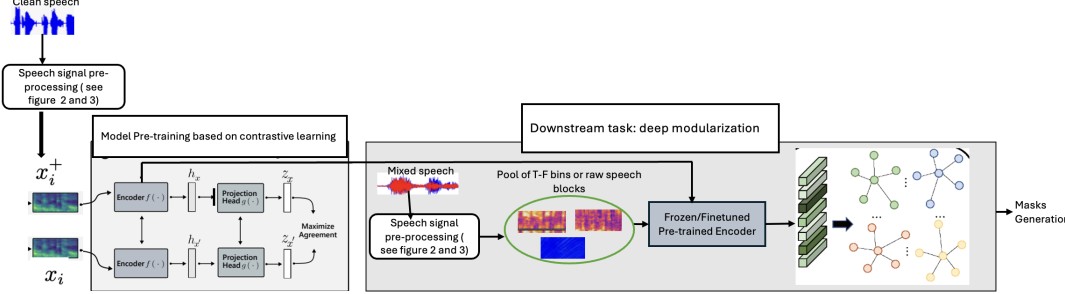

Figure 1: Overview of the proposed speech separation method based on deep modularization.

### 2.0.1 Generating T-F Bins

We begin with a pool of $n$ clean speech signals in the time domain, denoted as $x \in \mathbb{R}^T$. Each clean speech signal is augmented by mixing it with a randomly sampled noise excerpt, producing a noisy version set, $S_n$. To further augment the data, reverberation is applied to the noisy speech, creating a second version set, $S_{nr}$, containing speech signals with both noise and reverberation.

Both $S_n$ and $S_{nr}$ are downsampled to 8 kHz for computational efficiency. We then process the signals using the Short-Time Fourier Transform (STFT) with a 25 ms Hamming window and a 10 ms hop size. This setup strikes a balance between time and frequency resolution, capturing the audio's frequency content over short intervals. For a speech signal of duration $t$, this configuration generates a spectrogram of dimensions $128 \times 100t$, where 128 corresponds to the number of frequency bins, and $100t$ to the number of time frames.

From these spectrograms, we extract $3 \times 3$ patches of T-F bins with an overlap of 1 in both the time and frequency domains. This overlap ensures continuity and complete coverage of the spectrogram for further processing, providing detailed feature extraction.

We generate two sets of T-F bin patches:

- $\bar{S}_n$: $3 \times 3$ T-F bin patches for noisy speech signals.

- $\bar{S}_{nr}$: $3 \times 3$ T-F bin patches for noisy and reverberant speech signals.

Each $3 \times 3$ patch with overlap covers a total of 440 time-domain samples.

### 2.0.2 Generating Time-Domain Blocks as Input

Similar to the T-F bin generation process, we begin by processing clean speech signals $x \in \mathbb{R}^T$, and generate both noisy ($S_n$) and noisy + reverberant ($S_{nr}$) versions. These augmented signals are passed through an encoder, similar to the one proposed by (Subakan et al., 2021b), which produces an STFT-like representation $h \in \mathbb{R}^{F \times T}$.

The STFT-like representation is then divided into blocks of size 440 samples, with an 80-sample overlap along the time axis, resulting in a set of blocks $L \in \mathbb{R}^{F \times W \times N}$, where $W$ denotes the block length, and $N$ represents the number of blocks. The time-domain blocks are categorized into two sets:

- $\bar{X}_n$: Time-domain blocks for noisy speech signals.

- $\bar{X}_{nr}$: Time-domain blocks for noisy and reverberant speech signals.

### 2.1 Pre-trained Model

To extract meaningful representations from input speech, we employ a pre-trained model capable of processing either T-F bin patches or time-domain blocks. The model is trained on a large corpus of patches or speech blocks, enabling it to generate robust feature representations. These learned features serve as input to the subsequent clustering step.

We leverage contrastive learning (CL) to train the pre-trained model $f_\theta$, where the goal is to minimize the distance between similar objects (positive pairs) and maximize the distance between dissimilar objects (negative pairs) in the embedding space. Specifically, CL defines a representation function $f_\theta : x \mapsto \mathbb{R}^d$, mapping augmented inputs into $d$-dimensional vectors. The objective is to ensure that similar augmentations of the same input are close in the embedding space, while different inputs are pushed apart.

Typically, augmentations $(x, x^+)$ are generated by applying two distinct augmentation functions to the same input. In our case, these augmentations preserve critical features, such as speaker identity, while altering irrelevant aspects like noise and reverberation.

The contrastive learning model $f_\theta$, parameterized by $\theta$, is trained using either T-F bin patches or raw speech blocks as inputs. Positive pairs consist of two T-F bin patches or raw blocks from the same speaker, while

negative samples are patches or blocks from different speakers. For example, a speaker similarity function $\mathcal{T}$ randomly selects a positive pair $(x_{n_i}, x_{nr_i})$, where $x_{n_i} \in \bar{X}_n$ and $x_{nr_i} \in \bar{X}_{nr}$, ensuring both come from the same speaker.

For a batch of size $N$, given a positive pair $(x_{n_i}, x_{nr_i})$, the remaining $N - 2$ patches or blocks in the batch serve as negative samples. The contrastive learning objective is optimized via the following loss function:

$$\ell_n = -\log \frac{\exp(\text{sim}(z_{n_i}, z_{nr_i})/\tau)}{\sum\limits_{i'=1, i' \neq i}^{N} \exp(\text{sim}(z_{n_i}, z_{nr_{i'}})/\tau)}$$

Here, $\text{sim}(z_{n_i}, z_{nr_i})$ represents the similarity between the embeddings of the positive pair, and $\tau$ is a temperature parameter controlling the separation between positive and negative pairs.

Algorithm 1 outlines the training process of the pre-trained model using contrastive learning.

---

**Algorithm 1** Establishing the function $f_\theta$

1: **Initialize:** Data $\bar{X}_n$, $\bar{X}_{nr}$, $f_\theta(\cdot)$ (encoder), $g_\theta(\cdot)$ (projection head), and $\mathcal{T}$ (speaker similarity function)
2: **for** each minibatch of data $\{x_{n_i} : i \in N\}$ from $\bar{X}_n$ **do**
3:     **for** $i = 1$ to $N$ **do**
4:         Sample $x_{nr_i}$ from $\bar{X}_{nr}$ using $\mathcal{T}$
5:         $x_{n_i} \leftarrow 1^{\text{st}}$ augmentation of $x_{n_i}$
6:         $h_{n_i} = f_\theta(x_{n_i})$ {Extract embedding}
7:         $z_{n_i} = g_\theta(h_{n_i})$ {Project embedding}
8:         $x_{nr_i} \leftarrow 2^{\text{nd}}$ augmentation of $x_{nr_i}$
9:         $h_{nr_i} = f_\theta(x_{nr_i})$ {Extract embedding}
10:        $z_{nr_i} = g_\theta(h_{nr_i})$ {Project embedding}
11:     **end for**
12:     Define $\ell_n = -\log \frac{\exp(\text{sim}(z_{n_i}, z_{nr_i})/\tau)}{\sum\limits_{i'=1, i' \neq n}^{N} \exp(\text{sim}(z_{n_i}, z_{nr_{i'}})/\tau)}$
13:     $\ell = \frac{1}{N} \sum\limits_{n=1}^{N} \ell_n$ {Simplified SimCLR loss}
14:     Update encoder $f_\theta(\cdot)$ and projection head $g_\theta(\cdot)$ to minimize $\ell$ {Maximize agreement}
15: **end for**
16: **return** Encoder $f_\theta(\cdot)$

---

## 2.2 Deep Modularization for T-F Bin or Raw Speech Block Clustering

The goal is to use a deep modularization technique to cluster patches or raw speech blocks such that those dominated by a given speaker are clustered together. Deep modularization is a technique that integrates a graph clustering objective with a deep neural network (DNN). We start by defining a graph $G(V, E)$ where $V = (v_1, v_2, \cdots, v_n)$, with $|V| = n$, is the set of all patches or raw speech blocks generated from a mixed speech signal, as described in the speech pre-processing section. The set of edges is $E \subset V \times V$, with $|E| = m$, which connects the generated patches or raw speech blocks (subsequently referred to as "blocks"). The adjacency matrix of $G$ is denoted as $A$, where $A_{ij} = 1$ if $v_i, v_j \in E$, and 0 otherwise. The degree of $v_i$ is defined as $d_i = \sum_j^n A_{ij}$. Our objective is to generate a graph partition function $\mathcal{F} : V \to 1, \cdots, k$ that splits the set of patches or blocks $V$ into $k$ partitions, where $v_i = v_j : \mathcal{F}(v_j) = i$, given the patches or block attributes $\bar{F} \in R^{n \times d}$ learned from the pre-trained model.

To partition the vertices, we use the statistical approach known as modularity ($Q$) (Newman, 2006). Modularity measures the difference between the actual number of edges within partitions and the expected number of edges in equivalent randomized partitions (null networks). Formally, modularity is defined as:

$$Q = \text{Number of edges within partitions} - \text{Expected number of such edges}. \tag{1}$$

A high value of $Q$ indicates stronger similarity among members of the same partition. Therefore, the goal is to maximize $Q$. Let $g_i$ represent the community to which vertex $i$ belongs. Modularity ($Q$) is derived in (Newman, 2006) as:

$$Q = \frac{1}{2m} \sum_{ij} (A_{ij} - P_{ij}) \delta(g_i, g_j) \tag{2}$$

where $\delta(g_i, g_j)$ equals 1 if vertices $i$ and $j$ belong to the same partition and 0 otherwise. $P_{ij}$ is the expected number of edges between vertices $i$ and $j$, while $A_{ij}$ is the actual number of edges between them. If vertices $i$ and $j$ have degrees $d_i$ and $d_j$, respectively, then the expected degree of vertex $i$ is $\sum_j P_{ij} = d_i$. Hence, vertex $i$ and vertex $j$ are connected with probability $P_{ij} = \frac{d_i d_j}{2m}$ (Newman, 2006). Consequently, Equation 2 becomes:

$$Q = \frac{1}{2m} \sum_{ij} \left( A_{ij} - \frac{d_i d_j}{2m} \right) \delta(g_i, g_j) \tag{3}$$

Maximizing $Q$ is NP-hard (Brandes et al., 2006). To solve this, we define a partition assignment matrix $S \in R^{n \times k}$, where $n$ is the number of vertices. Each column of $S$ indexes a partition, meaning $S = s_1 \mid s_2 \mid \cdots \mid s_k$, with $S_{ij} = 1$ if vertex $i$ belongs to partition $j$, and 0 otherwise. The columns of $S$ are mutually orthogonal, as each row sums to 1, and $S$ satisfies the normalization $Tr(S^T S) = n$, where $Tr(\cdot)$ is the matrix trace. Based on this definition, $\delta(g_i, g_j) = \sum_{k=1}^{k} S_{ik} S_{jk}$, and Equation 3 becomes:

$$Q = \frac{1}{2m} \sum_{ij=1}^{n} \sum_{k=1}^{k} (A_{ij} - P_{ij}) S_{ik} S_{jk} = \frac{1}{2m} Tr(S^T B S) \tag{4}$$

where $B$ is the modularity matrix, defined as $B_{ij} = A_{ij} - P_{ij}$. By relaxing $S \in R^{n \times k}$, the optimal $S$ that maximizes $Q$ corresponds to the top $k$ eigenvectors of matrix $B$. Work by (Müller, 2023) enhances this objective by adding a regularization term to avoid trivial partitions, as shown in Equation 5. This regularization maintains consistency in community detection as the number of nodes increases (Müller, 2023):

$$L(S) = -\frac{1}{2m} Tr(S^T B S) + \frac{\sqrt{k}}{n} \left| \sum_i S_i^T \right|_F - 1 \tag{5}$$

Here, $|\cdot|_F$ is the Frobenius norm. The complexity of the modularity term $Tr(S^T B S)$ is $\mathcal{O}(n^2)$ per update of $L(S)$, making the training computationally expensive. To address this, (Müller, 2023) proposes to decompose $Tr(S^T B S)$ into sparse matrix-matrix multiplication and rank-one degree normalization, reducing the complexity to $\mathcal{O}(d^2 n)$:

$$L(S) = -\frac{1}{2m} Tr(S^T A S - S d^T d S) + \frac{\sqrt{k}}{n} \left| \sum_i S_i^T \right|_F - 1 \tag{6}$$

We adopt the objective in Equation 6 to modularize the features $\bar{F} \in R^{n \times d}$ learned via the pre-trained model. To optimize the cluster assignment, we adapt the deep neural network graph partitioning technique proposed in (Bianchi et al., 2020) and (Müller, 2023), where nodes of a graph are partitioned as follows:

$$\bar{F} = \text{GNN}(\tilde{A}, X, \theta_{\text{GNN}}) \tag{7}$$

$$S = \text{softmax}(\bar{F}) \tag{8}$$

Here, $\tilde{A} = D^{-\frac{1}{2}} A D^{-\frac{1}{2}}$, $X$ are the input features, $D$ is the diagonal matrix of degrees $d_1, \ldots, d_n$, and $A$ is the adjacency matrix. In Equation 7, node features $\bar{F}$ are learned via a graph neural network (GNN), and the assignment matrix $S$ is generated through a softmax function. In (Bianchi et al., 2020), the assignment matrix $S$ is established via a multilayer perceptron (MLP) with softmax at the output layer. In our case, we formulate the problem as:

$$\bar{F} = \text{Pre-M}(X, \theta_{\text{con}}) \tag{9}$$

$$S = \text{MOD}(\bar{F}, \theta_{\text{rnn}}) \tag{10}$$

Here, the feature matrix $\bar{F}$ is obtained via the pre-trained model (Pre-M). Since we extracted $3 \times 3$ patches and speech separation is typically performed over individual $1 \times 1$ T-F bins via masking, we apply a *Gaussian-weighted patch averaging* technique to compute the mean feature representation for each $1 \times 1$ T-F bin. To preserve local dependencies, we restrict the averaging to the three closest neighboring T-F bins, i.e., $|C| = 3$, in both the time and frequency domains. For each feature vector $\bar{f}_j \in \bar{F}$, the Gaussian-weighted averaged feature is computed as:

$$\bar{f}_{\text{avg},j} = \frac{\sum_{i \in C} w_i \cdot \bar{f}_i}{\sum_{i \in C} w_i}$$

where $C$ consists of the central T-F bin and its two closest neighbors. The weights $w_i$ are assigned using a 2D Gaussian kernel, emphasizing the importance of proximity within the local T-F region to retain crucial local information.

The averaged feature $\bar{f}_{\text{avg},j}$ provides a compact and locally representative feature that captures the most relevant information from the neighboring T-F bins and serves as input to the BLSTM model.

The BLSTM, as proposed in Huang et al. (2022), takes the averaged patch feature and assigns it to the $j$-th column of the cluster assignment matrix $S$. The final cluster assignments are established using a softmax at the output layer.

To optimize the assignment $S$, we use the loss function described in Equation 6, effectively training a DNN model using a graph clustering objective but with averaged patch features. This enhances the robustness of the learned features by reducing noise and capturing more meaningful representations for each patch.

For *speech blocks*, no further averaging or processing is applied, and the features $\bar{F}$ are used directly for clustering or downstream tasks.

## 2.3 Adjacency matrix

To construct the adjacency matrix $A$, we first measure the similarity between each T-F bin or block $i$ and all other nodes in the feature space. This similarity is computed using the inner product:

$$e_{ij} = \bar{f}_i^T \bar{f}_j$$

where $i, j = 1, 2, \ldots, n$, and $\bar{f}_i$ and $\bar{f}_j \in \bar{F}$ are feature vectors obtained from the pre-trained model (Pre-M). The higher the inner product, the more similar the two feature vectors are.

Next, to form the edges of the graph, we select a similarity threshold $\theta$. If $e_{ij} < \theta$, we remove the edge between nodes $i$ and $j$, thereby limiting the graph to meaningful edges. The adjacency matrix is defined as:

$$A_{ij} = \begin{cases} 1, & \text{if } e_{ij} \geq \theta \text{ (edge exists)} \\ 0, & \text{if } e_{ij} < \theta \text{ (no edge)} \end{cases}$$

Choosing an optimal threshold $\theta$ is crucial for constructing a meaningful graph. A high value of $\theta$ ensures only highly similar nodes are connected, but this can exclude meaningful relationships. On the other hand, a low $\theta$ results in a dense graph filled with weaker, uninformative edges, which increases computation time during clustering.

To find the optimal $\theta$, we experimented on multiple datasets, varying the threshold values and observing both modularity and the number of clusters generated. Modularity measures the quality of clusters, with higher values indicating better-defined clusters.

Figure 2 shows how modularity and the number of clusters change with varying $\theta$ on the WSJ0-5mix test dataset. For visualization purposes, modularity values are normalized (multiplied by 100) and the number of clusters is scaled by 10. As shown in the figure, increasing $\theta$ boosts modularity but risks creating singleton partitions. Decreasing $\theta$, however, increases unnecessary clusters and lowers modularity.

After careful evaluation, we selected $\theta = 0.3$ as the optimal threshold, which was used consistently across all our experiments.

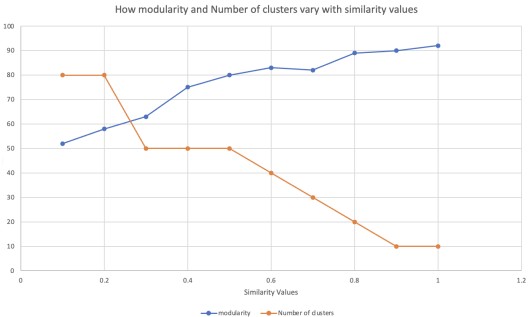

Figure 2: Modularity and number of clusters as a function of the similarity threshold $\theta$.

## 3  Clean source signal estimation

We use the downstream model to classify patches or blocks into $k$ partitions. Once the model generates the probability distribution via softmax, we use this distribution to generate soft masks. Each mask is a continuous vector composed of values between 0 and 1. A value close to 0 indicates minimal contribution from a particular T-F bin in the corresponding cluster, while a value closer to 1 signifies dominance of the source in that bin. These masks are based on the assumption of sparsity and orthogonality of the sources in the domain where they are computed. This assumption implies that, within any given T-F bin, the dominant source is the primary contributor, though partial contributions from other sources are possible. As a result, each T-F bin is assumed to correspond to a weighted combination of sources.

Once the soft masks are constructed, they are applied to the mixed signal, isolating $k$ estimated clean sources. For speech signals that have been transformed into their Short-Time Fourier Transform (STFT) representation, the masks are directly applied to the STFT spectrogram to estimate the spectrograms of the individual clean speech signals. The inverse STFT is then applied to each estimated spectrogram to reconstruct the corresponding clean speech signals in the time domain.

In the case of raw time-domain signals, the masks are applied to STFT-like representations generated by the encoder, which learns features analogous to T-F bins. The masked representations are then passed through the decoder (a transposed version of the encoder) to generate the estimated clean speech signals in the time domain.

For phase reconstruction in the STFT domain, we adopt the technique proposed by Wang (2018), which jointly reconstructs the phase of all sources in each mixture by leveraging their estimated magnitudes and the noisy phase information. This is achieved using the Multiple Input Spectrogram Inversion (MISI) algorithm (Gunawan 2010), which iteratively refines the phase to ensure coherence across the estimated signals.

## 4 Experimental setup

**Pre-training Dataset:** To pre-train the two model variants—one using patches as input and the other using raw speech blocks—we utilized the widely known Wall Street Journal (WSJ0) corpus (Paul & Baker, 1992). This dataset was recorded using a close-talk microphone, ensuring it is free from both reverberation and noise. We used 30 hours of speech data from the *si_tr_s* subset to train the models.

To generate augmented versions of the speech data, we first added noise and then convolved the noisy speech with impulse responses. For noise addition, we used the noise dataset from (Wichern et al., 2019), which contains recordings captured using a binaural microphone in various locations around the San Francisco Bay area. Each speech sample was mixed with a randomly selected noise excerpt from this dataset, ensuring that the signal-to-noise ratio (SNR) between the speech and noise in each mixture was less than 2. For reverberation, we convolved the noisy speech with impulse responses provided by (Maciejewski et al., 2020), specifically using the medium reverberation settings. These impulse responses simulate various acoustic environments, with reverberation times ($T_{60}$) uniformly sampled between 0.2 and 0.6 seconds. From the augmented speech data, we extracted patches and raw speech blocks as described in the speech pre-processing section. The 30 hours of speech data generated a dataset of over 59 million patches or speech blocks.

**Pre-trained Model:** We trained a model known as TFBlockNet, based on a convolutional neural network (CNN) architecture designed for embedding two-dimensional inputs of patches or raw speech blocks. The architecture is built around multiple layers of Mobile Inverted Bottleneck Convolutional Blocks (MBConv), which are recognized for their efficiency and ability to capture complex patterns with fewer parameters. The network concludes with a global pooling layer, followed by a fully connected projection layer (MLP). This MLP is responsible for projecting the high-dimensional feature maps into a lower-dimensional embedding space suitable for contrastive learning. The final embedding vector is set to 128 dimensions, optimizing the model for clustering. The architecture is summarized in Table 1 below.

Table 1: Breakdown of the TFBlockNet Architecture by Stages

| Stage | Operator | #Channels | #Layers |
|---|---|---|---|
| 1 | Conv3x3 | 32 | 1 |
| 2 | MBConv1, k3x3 | 16 | 1 |
| 3 | MBConv6, k3x3 | 24 | 2 |
| 4 | MBConv6, k5x5 | 40 | 2 |
| 5 | MBConv6, k3x3 | 80 | 3 |
| 6 | MBConv6, k5x5 | 112 | 3 |
| 7 | MBConv6, k5x5 | 192 | 4 |
| 8 | MBConv6, k3x3 | 320 | 1 |
| 9 | Conv1x1 & Pooling & FC | 1280 | 1 |

**Pre-training Configuration:** Two variants of the model were pre-trained—one on batches and the other on raw speech blocks—using the Adam optimizer. A cyclical learning rate schedule (Smith, 2017) was employed, with the learning rate cycling between $1 \times 10^{-4}$ and $1 \times 10^{-1}$. This triangular policy, cycling the learning rate over 10,000 steps, allows for faster convergence and helps avoid overfitting by continuously adjusting the learning rate throughout training. Each model was trained for 2 million steps on a single NVIDIA V100 GPU with a batch size of 512, ensuring efficient utilization of computational resources while maintaining robust feature learning.

**Speech Separation Dataset:** To evaluate the quality of the separated speech resulting from the proposed technique, we used several standard benchmark datasets: WSJ0-2mix, WSJ0-3mix (Hershey et al., 2016), WSJ0-4mix, WSJ0-5mix (Nachmani et al., 2020), Libri5Mix, and Libri10Mix (Dovrat et al., 2021). These datasets contain mixtures of 2, 3, 4, and 5 speakers (WSJ0-based), or 5 and 10 speakers (LibriMix-based), respectively.

The WSJ0-based datasets (WSJ0-2mix, WSJ0-3mix, WSJ0-4mix, and WSJ0-5mix) were generated from the WSJ0 corpus by mixing clean speech utterances from multiple speakers. Random gains were applied to achieve relative levels between the speakers, ranging from 0 to 5 dB, to simulate real-world conditions. Each dataset consists of 30 hours of training data, 10 hours of validation data, and 5 hours of test data. The training and validation sets share common speakers, while the test set contains unseen speakers, providing a robust evaluation of the model's generalization ability.

The LibriMix-based datasets, Libri5Mix and Libri10Mix, were created from the LibriSpeech corpus (Panayotov et al., 2015), and contain more complex mixtures involving 5 and 10 speakers, respectively. These datasets were generated without background noise, with the resulting mixtures having SNRs (signal-to-noise ratios) that are normally distributed with a mean of 0 dB and a standard deviation of 4.1 dB (Dovrat et al., 2021). This variety of mixtures represents different levels of complexity in speech separation tasks.

Both the WSJ0 and LibriMix datasets assume an anechoic environment, where speech is recorded in ideal, noiseless conditions. However, this assumption is unrealistic for practical speech separation, where speech is often recorded with distant microphones, introducing noise and reverberation. To address this, we also evaluated the performance of the proposed technique using the WHAM! (Wichern et al., 2019) and WHAMR! (Maciejewski et al., 2020) datasets. These datasets are derived from WSJ0-2mix by adding environmental noise (WHAM!) and both noise and reverberation (WHAMR!), thereby creating more challenging and realistic conditions for speech separation.

For all datasets, the test set was used to evaluate speech separation performance. The patches or raw blocks generated from the test speech mixtures were processed by the relevant pre-trained model to generate embeddings, depending on the type of input the model was trained on (i.e., patches or raw speech blocks).

**Clustering:** We explored two configurations for clustering the features generated by the pre-trained models. In the first configuration, we trained a modularization module (MOD) (see Equation 6, the clustering loss function) on top of a frozen encoder to partition the patches or raw speech blocks. In the second configuration, we fine-tuned the pre-trained encoder during clustering, optimizing the encoder in an end-to-end differentiable manner while still using Equation 6 as the loss function. These two configurations allowed us to explore whether fine-tuning the encoder would improve clustering performance or if a frozen encoder could generalize effectively.

We trained four different MOD configurations:

1. MOD1: Clustering of patches using a frozen TFBlockNet pre-trained on patches.

2. MOD2: Clustering of raw speech blocks using a frozen TFBlockNet pre-trained on raw speech blocks.

3. MOD3: Clustering of patches with a fine-tuned TFBlockNet pre-trained on patches.

4. MOD4: Clustering of raw speech blocks with a fine-tuned TFBlockNet pre-trained on raw speech blocks.

Each model was trained using the Adam optimizer with a cyclical learning rate schedule (Smith, 2017), where the learning rate ranged from $1 \times 10^{-4}$ to $1 \times 10^{-1}$. Training was conducted on a single NVIDIA V100 GPU for 2 million steps, with a batch size of 512 T-F bins or raw speech blocks per step. The number of clusters, $k$, was varied based on the dataset, e.g., $k = 3$ for the WSJ0-3mix test dataset.

**Evaluation Metrics:** We evaluated the performance of our speech separation models using several objective metrics, including both traditional measures and newer, deep learning-based metrics. The traditional metrics include Short-Time Objective Intelligibility (STOI) (Taal et al., 2011), Perceptual Evaluation of Speech Quality (PESQ) (Rix et al., 2001), SI-SNR improvement (SI-SNRi), and Signal-to-Distortion Ratio improvement (SDRi).

STOI measures the intelligibility of the separated speech, particularly in noisy environments. PESQ provides a perceptual evaluation of speech quality, correlating well with human listening scores. SI-SNRi evaluates the improvement in the signal-to-interference ratio, reflecting how effectively the model separates the target speech from interfering sources. SDRi assesses the improvement in overall signal quality after separation.

In addition to these traditional metrics, we utilized the Deep Noise Suppression Mean Opinion Score (DNS-MOS) (Reddy et al., 2021), a reference-free metric that evaluates perceptual speech quality. DNSMOS is based on a deep neural network (DNN) trained on human ratings and follows an online framework for listening experiments, adhering to the ITU-T P.808 standard.

We also incorporated the SIG, BAK, and OVRL scores from the non-intrusive speech quality assessment model DNSMOS P.835 (Reddy et al., 2022). These scores evaluate different aspects of speech quality:

- **SIG:** Assesses the quality of the target speech signal.

- **BAK:** Measures the suppression of background noise.

- **OVRL:** Provides an overall quality score.

## 5 Results

### 5.1 Quality of Clusters

We first evaluated which of the four model configurations (MOD1, MOD2, MOD3, MOD4) produces better clusters (partitions). To measure cluster quality, we used the graph-based cluster evaluation metrics proposed in (Yang & Leskovec, 2012), focusing on metrics that quantify how well a partition is separated from the rest, i.e., the number of edges connecting a given partition to other partitions. A high-quality partition should have few edges pointing outside. The most relevant metrics for our study are graph modularity and conductance.

**Cluster Conductance (C)** is defined as $C = \frac{c_s}{2m_s + c_s}$, where $S$ is a partition, $m_s$ is the number of edges within $S$ (i.e., $m_s = \{(u,v) \in E \mid u \in S, v \in S\}$), and $c_s$ is the number of edges on the boundary of $S$ (i.e., $c_s = \{(u,v) \in E \mid u \in S, v \notin S\}$). Conductance measures the fraction of edges pointing outside a partition, with lower values indicating better quality.

**Graph Modularity ($\mathcal{Q}$)** is defined as $\mathcal{Q} = \frac{1}{4}(m_s - E(m_s))$, where $E(m_s)$ is the expected value of $m_s$. Higher modularity indicates better partition quality.

The results of our evaluation are shown in Table 2. Comparing similarly trained models across all datasets, we found that encoders trained on raw speech blocks produced higher-quality clusters than those trained on T-F bin patches. This suggests that for this setup, raw speech features capture more relevant speech information compared to STFT-transformed features. Additionally, fine-tuning the encoder significantly improved cluster quality compared to using a frozen encoder.

Table 2: Results of conductance C and modularity Q when using different input configurations and different mixtures. Here the values of C and Q have been multiplied by 100.

| Model Configuration | C($\downarrow$) | $\mathcal{Q}$($\uparrow$) |
|---|---|---|
| **WSJ0-3mix test-dataset** | | |
| MOD1 | 14.9 | 85.1 |
| MOD2 | 14.4 | 86.7 |
| MOD3 | 14.1 | 88.5 |
| MOD4 | 13.9 | 88.9 |
| **WSJ0-4mix test-dataset** | | |
| MOD1 | 16.2 | 85.3 |
| MOD2 | 15.8 | 86.9 |
| MOD3 | 15.1 | 87.2 |
| MOD4 | 14.6 | 87.6 |
| **WSJ0-5mix test-dataset** | | |
| MOD1 | 19.0 | 76.6 |
| MOD2 | 18.4 | 78.5 |
| MOD3 | 18.0 | 78.8 |
| MOD4 | 17.7 | 79.6 |

## 5.2 Evaluation of Speech Separation Quality

We evaluated the performance of four speech separation models (MOD1, MOD2, MOD3, and MOD4) using the WSJ0-2mix, WSJ0-3mix, and WSJ0-4mix test datasets. The results, summarized in Table 3, show that MOD4 consistently achieved the highest performance across all metrics, followed by MOD2, irrespective of the number of speakers in the mixture. Specifically, MOD4 demonstrated significant improvement in SDRi (Signal-to-Distortion Ratio improvement), SI-SNRi (Scale-Invariant Signal-to-Noise Ratio improvement), and STOI (Short-Time Objective Intelligibility), outperforming the other models across all test sets.

Interestingly, this contrasts with the clustering quality evaluation, where MOD4 also produced the best clusters, but MOD3 outperformed MOD2 in cluster quality. The lower performance of MOD3 compared to MOD2 in separation quality can be attributed to phase reconstruction issues that arise when using STFT-transformed features. The phase reconstruction process, essential for high-fidelity speech reconstruction, is more challenging with MOD3's approach, leading to suboptimal separation performance.

Across the three datasets, the superior performance of MOD4 and MOD2 is consistent with models trained on raw speech blocks for both pre-training and downstream tasks. This indicates that raw speech blocks capture more relevant and meaningful features for speech separation compared to models trained with STFT-transformed patches (T-F bins). The inferior performance of the T-F bin-based models (MOD1 and MOD3) likely stems from the phase reconstruction difficulties inherent in this approach, which is further exacerbated in the separation process.

Table 3: Speech separation results when the two variants of inputs are used

| Model | SDRi($\uparrow$) | SI-SNRi($\uparrow$) | STOI($\uparrow$) | PESQ ($\uparrow$) | DNSMOS ($\uparrow$) | SIG ($\uparrow$) | BAK ($\uparrow$) | OVRL ($\uparrow$) |
|---|---|---|---|---|---|---|---|---|
| WSJ0-2mix test-dataset | | | | | | | | |
| MOD1 | 16.0 | 15.8 | 0.8969 | 3.93 | 3.98 | 3.96 | 4.11 | 4.01 |
| MOD2 | 21.9 | 21.6 | 0.9146 | 4.04 | 4.14 | 4.09 | 4.17 | 4.16 |
| MOD3 | 16.2 | 16.0 | 0.9054 | 3.98 | 4.02 | 4.00 | 4.13 | 4.07 |
| MOD4 | **22.6** | **22.3** | **0.9346** | **4.08** | **4.17** | **4.11** | **4.23** | **4.20** |
| WSJ0-3mix test-dataset | | | | | | | | |
| MOD1 | 15.2 | 14.8 | 0.8702 | 3.76 | 3.86 | 3.88 | 3.91 | 3.89 |
| MOD2 | 19.7 | 19.5 | 0.9051 | 4.01 | 4.05 | 4.01 | 4.08 | 4.11 |
| MOD3 | 15.5 | 15.2 | 0.8837 | 3.86 | 3.98 | 3.91 | 4.07 | 4.01 |
| MOD4 | **21.8** | **21.3** | **0.9150** | **4.04** | **4.10** | **4.07** | **4.14** | **4.09** |
| WSJ0-4mix test-dataset | | | | | | | | |
| MOD1 | 14.4 | 14.0 | 0.8414 | 3.52 | 3.67 | 3.75 | 3.81 | 3.78 |
| MOD2 | 17.4 | 17.0 | 0.8896 | 3.92 | 3.97 | 3.94 | 4.01 | 4.03 |
| MOD3 | 15.3 | 15.0 | 0.8522 | 3.70 | 3.84 | 3.87 | 4.03 | 3.98 |
| MOD4 | **20.9** | **20.5** | **0.9011** | **3.98** | **4.02** | **4.00** | **4.08** | **4.04** |

## 5.3 Comparison with Other Speech Separation Tools in Few-Source Mixtures ($n \leq 3$)

We evaluated the performance of the proposed speech separation technique (MOD4) against several state-of-the-art models on the WSJ0-2mix and WSJ0-3mix test datasets. The results, summarized in Table 4, demonstrate that MOD4 consistently achieves competitive results. On the WSJ0-2mix dataset, MOD4 attained an SI-SNRi of 22.6, trailing MossFormer2 by 1.5 points, and an SDRi of 22.9, which is only 0.6 points lower than the top-performing TF-GridNet. These results indicate that MOD4, despite not relying on parallel datasets, is highly competitive with fully supervised methods.

On the WSJ0-3mix dataset, MOD4's robustness and scalability to higher-source mixtures become evident. While MossFormer2's SI-SNRi dropped by 1.9 points between the WSJ0-2mix and WSJ0-3mix tasks, MOD4 only experienced a minor drop of 0.8 points. This ability to maintain performance across more complex mixtures highlights MOD4's generalization capability, with MOD4 achieving an SI-SNRi of 21.8, only 0.4 points behind MossFormer2, showing strong scalability as the number of sources increases.

Furthermore, MOD1 and MOD3 outperformed $\mathcal{L}MISI - 5^{Enh3}$, a supervised STFT-based method, which uses non-negative masking with MISI for phase reconstruction. The superior performance of MOD1 and MOD3 suggests that the proposed models' ability to more accurately predict the magnitude of the source signal results in higher-quality separation, even in the absence of parallel data. This highlights the strength of raw speech block models over STFT-based models for speech separation tasks.

In conclusion, the results show that the proposed technique (MOD4) generalizes well to more complex mixtures and performs competitively against top supervised models, despite not relying on parallel datasets, making it an attractive choice for real-world applications where parallel data is limited.

Table 4: Comparing the results of the proposed technique with other state of the art speech separation tools.

| WSJ0-2mix test-dataset | | |
|---|---|---|
| Model | SI-SNRi | SDRi |
| TF-GridNet (Wang et al., 2023) | 23.5 | **23.6** |
| MossFormer2 (Zhao et al., 2024) | **24.1** | - |
| SepFormer (Subakan et al., 2021b) | 20.4 | 20.5 |
| SepFormer+DM (Subakan et al., 2021b) | 22.3 | 22.4 |
| Wavesplit (Zeghidour & Grangier, 2021b) | 21.0 | 21.2 |
| Wavesplit+DM (Zeghidour & Grangier, 2021b) | 22.2 | 22.3 |
| DeepCASA (Liu & Wang, 2019b) | 17.7 | 18.0 |
| ConvTasnet (Luo & Mesgarani, 2019b) | 15.3 | 15.6 |
| MixIT(on 8KHz test data) (Wisdom et al., 2020a) | 17.0 | - |
| $\mathcal{L}_{MISI-5}^{Enh3}$(Wang et al., 2019) | 15.3 | 15.6 |
| MOD1 | 15.8 | 16.0 |
| MOD2 | 21.9 | 22.1 |
| MOD3 | 16.0 | 16.2 |
| MOD4 | 22.6 | 22.9 |
| WSJ0-3mix test-dataset | | |
| MossFormer2 | **22.2** | - |
| SepFormer | 17.6 | 17.9 |
| SepFormer+DM | 19.5 | 19.7 |
| Wavesplit | 17.3 | 17.6 |
| Wavesplit+DM | 17.8 | 18.1 |
| ConvTasnet | 12.7 | 13.1 |
| $\mathcal{L}_{MISI-5}^{Enh3}$(Wang et al., 2019) | 12.1 | 12.5 |
| MOD1 | 14.8 | 15.2 |
| MOD2 | 19.7 | 19.7 |
| MOD3 | 15.2 | 15.5 |
| MOD4 | **21.8** | **21.9** |

Table 5: Performance of the proposed technique on high source mixtures as compared to other tools that can perform high source mixtures separation.

| WSJ0-5mix test-dataset | |
|---|---|
| Model | SDRi |
| ConvTasNet (Luo & Mesgarani, 2019a) | 6.8 |
| DPRNN (Luo et al., 2020) | 8.6 |
| MulCat (Nachmani et al., 2020) | 10.6 |
| Hungarian (Dovrat et al., 2021) | 13.2 |
| MOD1 | 8.4 |
| MOD2 | 12.3 |
| MOD3 | 9.9 |
| MOD4 | **13.4** |
| Libri5Mix test-dataset | |
| SinkPIT (Tachibana, 2021) | 9.4 |
| MulCat (Nachmani et al., 2020) | 10.8 |
| Hungarian (Dovrat et al., 2021) | 12.7 |
| MOD1 | 8.6 |
| MOD2 | 11.5 |
| MOD3 | 10.1 |
| MOD4 | **13.0** |
| Libri10Mix test-dataset | |
| SinkPIT (Tachibana, 2021) | 6.8 |
| MulCat (Nachmani et al., 2020) | 4.8 |
| Hungarian (Dovrat et al., 2021) | 7.8 |
| MOD1 | 6.9 |
| MOD2 | 8.1 |
| MOD3 | 7.3 |
| MOD4 | **9.8** |

## 5.4 Comparison with Other Speech Separation Tools in High-Source Mixtures ($n \geq 5$)

We evaluated the performance of the proposed technique on mixtures with a large number of sources. The results, presented in Table 5, demonstrate that MOD4 achieves superior performance across the SDRi metric, outperforming existing tools by 0.2, 0.3, and 2.0 points on the WSJ0-5mix, Libri5Mix, and Libri10Mix datasets, respectively. This consistent performance highlights the ability of the proposed technique to scale effectively to high-source mixtures while maintaining high-quality separation.

Specifically, in the WSJ0-5mix test set, MOD4 achieved the highest SDRi score of 13.4, surpassing the previous top-performing Hungarian algorithm by 0.2 points. Similarly, in the Libri5Mix test set, MOD4 obtained an SDRi score of 13.0, outperforming Hungarian by 0.3 points. Notably, the most significant improvement was observed in the Libri10Mix test set, where MOD4 outperformed Hungarian by 2.0 points, further demonstrating its scalability in handling increasingly complex mixtures.

These results suggest that MOD4's architecture and underlying techniques are well-suited for separating a larger number of sources, surpassing several state-of-the-art methods. This scalability is especially important for real-world applications, where mixtures of many sources are common, and maintaining high separation quality becomes increasingly challenging.

### 5.5 Results on WHAM! and WHAMR! Datasets

We also assessed the proposed technique on two datasets that include noise and reverberation: WHAM! and WHAMR!. The results, summarized in Table 6, show that MOD4 achieved the best performance compared to other state-of-the-art methods on both datasets. On the WHAM! dataset, which requires simultaneous speech separation and denoising, the model was able to effectively handle the additional noise partition, demonstrating its robustness in noisy environments. Similarly, on the WHAMR! dataset, where the model must handle denoising, dereverberation, and speech separation simultaneously, MOD4 outperformed competing models across both the SI-SNRi and SDRi metrics.

These results highlight the proposed technique's ability to generalize well to real-world conditions involving both noise and reverberation, maintaining high separation quality under challenging scenarios. Compared to models like SepFormer and Wavesplit with dynamic masking (DM), MOD4's consistent performance across both noise and reverberation tasks further demonstrates its versatility and robustness in handling complex acoustic conditions.

Table 6: Comparing the results of the proposed technique with other state-of-the-art speech separation tools on WHAM and WHARM test-datasets.

| WHAM test-dataset | | |
|---|---|---|
| Model | SI-SNRi | SDRi |
| SepFormer (Subakan et al., 2021b) | 14.7 | 15.1 |
| SepFormer+DM (Subakan et al., 2021b) | 16.4 | 16.7 |
| Wavesplit+DM (Zeghidour & Grangier, 2021b) | 16.0 | 16.5 |
| ConvTasnet (Luo & Mesgarani, 2019b) | 12.7 | - |
| MOD1 | 13.6 | 13.9 |
| MOD2 | 16.4 | 16.8 |
| MOD3 | 13.8 | 14.0 |
| MOD4 | 17.2 | 17.4 |
| WHARM test-dataset | | |
| SepFormer | 11.4 | 10.4 |
| SepFormer+DM | 14.0 | 13.0 |
| Wavesplit+DM | 13.2 | 12.2 |
| ConvTasnet | 8.3 | - |
| BiLSTM Tasnet | 9.2 | - |
| MOD1 | 12.9 | 13.1 |
| MOD2 | 14.1 | 14.4 |
| MOD3 | 13.2 | 13.5 |
| MOD4 | 15.0 | 15.4 |

## 6 Ablation

### 6.1 $k$-Means vs. Deep Modularization

In this experiment, we replaced deep modularization with classical $k$-means clustering to generate partitions, setting $k = 2$ for the WSJ0-2mix test dataset and $k = 3$ for the WSJ0-3mix test dataset. Specifically, after extracting T-F bin patches or raw speech block features using a frozen pre-trained encoder, we applied $k$-means to cluster these features and generate partitions, which were subsequently used for mask generation. We evaluated $k$-means in two configurations:

1. $k$-means (raw speech blocks): Clustering was applied to features extracted from raw speech blocks.

2. $k$-means (patches): Clustering was applied to features extracted from patches.

The mask generation and speech reconstruction processes remained identical to those outlined in Section 3. We compared the performance of $k$-means clustering with that of deep modularization in terms of speech separation quality. Since $k$-means uses a frozen pre-trained encoder for feature extraction, we benchmarked its performance against MOD1 and MOD2, which also leverage a frozen encoder.

As shown in Table 7, deep modularization consistently outperformed $k$-means across all test datasets. These results suggest that deep modularization is more effective at capturing the relationships between T-F bins patches or raw speech blocks, resulting in superior mask generation and better speech separation quality compared to the classical $k$-means approach.

Table 7: Comparing the results of the proposed technique with other state of the art speech separation tools.

| WSJ0-2mix test-dataset | | |
|---|---|---|
| Model | SI-SNRi | SDRi |
| MOD1 | 15.8 | 16.0 |
| MOD2 | 21.9 | 22.1 |
| $k$-means(raw speech blocks) | 17.5 | 17.8 |
| $k$-means(T-F bin) | 12.3 | 12.4 |
| WSJ0-3mix test-dataset | | |
| MOD1 | 14.8 | 15.2 |
| MOD2 | 19.7 | 19.7 |
| $k$-means(raw speech blocks) | 15.5 | 15.9 |
| $k$-means(T-F bin) | 11.4 | 11.9 |

## 7    Conclusion

This work introduces an unsupervised technique for speech separation, leveraging a pre-trained model to generate input features, which are subsequently used in downstream tasks. In the downstream task, we integrate deep neural networks with graph clustering objectives to create clusters of spectrogram points or raw speech blocks, grouping T-F bins or blocks dominated by the same speaker together. Extensive experiments across multiple datasets demonstrate that the proposed method achieves state-of-the-art performance in both clean and noisy environments.

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

## Impact Statement

"This paper presents work whose goal is to advance the field of Machine Learning. There are many potential societal consequences of our work, none which we feel must be specifically highlighted here.

# A   Appendix

