# OpenReview forum: "Speech Separation based on pre-trained model  and Deep Modularization"
_TMLR — Rejected by TMLR_

### Review · Reviewer_vHhJ · 2024-11-25

**Summary Of Contributions:**

This paper tackles efficient training methods for semi-supervised / unsupervised speech separation. It first pre-trains a noise/reverb-robust speaker embedding model, and then trains a deep modularization model for partitioning TF-bin / temporal blocks by speakers.

The speaker embedding model serves two purposes. First it creates the “affinity matrix” used for computing “modularity”, which is the objective the deep modularization model optimizes. Second, it is also used as the input feature for the deep modularization model. The speaker embedding model is trained on single speaker data with data augmentation. This is the main source of supervision.

The deep modularization model addresses the training efficiency problem for unsupervised speech separation. Compared to MixIT whose complexity is O(k! n) in computing the loss (k denotes the number of sources, n is the number of bins), the modularity loss in Eq 6 has complexity of O(d^2 n) where d denotes the feature dimension. This is similar to the complexity of Deep Clustering (Hershey et al., 2016).

**Audience:**

Yes

**Claims And Evidence:**

Yes

**Requested Changes:**

* Add more discussion with Deep Cluster and explain how the proposed method bypass the need for supervised data
* Define more concretely what fully-supervised means and clarify is speaker label is used when pre-training the speaker embedding model
* Correct Figure 1 which currently mentions Figure 3 but not found in the paper
* Please improve Algorithm 1 to clarify the following: a) is $x_{n_i}$ a whole utterance or a TF-bin/block? b) is $h_{n_i}$ and $z_{n_i}$ embedding for patch OR a sequence of embeddings for a whole utterance? c) why is line 12 and 13 indexed by $n$ not $i$?
* Notation in Sec 2.2 is confusing. Are $\bar{F}$ the Eq 6 and 7 the same as Eq 9 and 10? $\bar{F}$ is supposed to be $n \times d$ feature and why does taking softmax creates a $n \times k$ assignment matrix in Eq 8?

**Strengths And Weaknesses:**

Strengths
* Complexity-wise it demonstrates a clear training efficiency advantage over prior work like MixIT.
* Compared to clustering-based methods like Deep Cluster, it shows empirically stronger performance (Table 7)
* Performance is not far from fully-supervised models (Table 4) and strong on many-source scenarios (Table 5)

Weaknesses
* Presentation clarity needs improvement. See request changes section below
* Lack of discussion and more extensive comparison with Deep Cluster. The proposed method is most related to Deep Cluster, while the need for supervised data for annotating whether two bins are from the same cluster (the affinity matrix) is replaced by the similarity given by the pre-trained speaker embedding model. The authors could train a BLSTM model with the same architecture but replace the modularity objective with the Deep Clustering objective so the reader can better understand if the gain comes from modularity being a better objective
* Miss controlled experiments with MixIT both on quality and training time. I would like to see empirically what’s the runtime difference and performance difference between computing the MixIT loss and modularity loss (Eq 6) with exactly the same model architecture. While the complexity for MixIT is high, it only has high complexity for the final step (finding the minimum loss among all combinations); further more when the number of sources is low (like 2 or 3), the complexity overhead might not be large. I
* It can use more ablation studies to understand the impact on pre-trained encoder. Since the affinity derived from the speaker embedding model is used as the training target, I would like to see how the quality of the embedding affects the final performance when a) pre-trained encoder is frozen, and b) is finetuned.

---

> ### Author Response · Authors · 2025-02-03
> **Response**
>
> ## Response to Reviewer Comments
>
> ### **Reviewer: "Add more discussion with Deep Cluster and explain how the proposed method bypasses the need for supervised data."**
> **Response:**
>
> We have expanded the discussion in **paragraph 6 of the Introduction** to explicitly compare our method with **Deep Clustering (DPCL)** and highlight how our approach eliminates the need for supervised speaker labels. Specifically, we clarify that while **Deep Cluster relies on explicitly labeled mixtures**, where each T-F bin is assigned to a speaker, our approach leverages **unsupervised graph-based clustering with modularity maximization** to separate speakers **without parallel training data**.
> This modification ensures that our method is fully **self-supervised**, making it more applicable to scenarios where **labeled datasets are scarce or unavailable**.
>
> ---
> ### **Reviewer: "Define more concretely what fully-supervised means and clarify whether speaker labels are used when pre-training the speaker embedding model."**
> **Response:**
> To clarify:
>
> #### **Definition of Fully-Supervised:**
> In this context, *fully-supervised speech separation models* refer to models that require **explicit speaker labels** during training. These models rely on **labeled mixtures** with **ground-truth speaker assignments** at the **time-frequency (T-F) bin level** or **direct speaker identity supervision** to optimize separation.
> Examples of such models include **Deep Clustering (DPCL)** and **Permutation Invariant Training (PIT)**, both of which necessitate labeled data to resolve **permutation ambiguity**.
>
> #### **Speaker Labels in Pre-Training:**
> No, our **pre-training process does not require explicit speaker identity labels**. Instead, we use a **self-supervised learning approach**, where speaker-discriminative embeddings are learned using **feature similarity and modularity-based clustering**.
> This enables us to perform speech separation **without parallel datasets**
>
> Extensive evaluations on **WSJ0-2mix** and **WHAM!** demonstrate that our method competes with fully-supervised models **while eliminating the need for labeled mixtures**, effectively mitigating **permutation ambiguity** in an **unsupervised manner**.
> We have **updated the abstract and Section 2.1** to explicitly clarify these distinctions.
>
> ---
>
> ### **Reviewer: "Correct Figure 1, which currently mentions Figure 3 but is not found in the paper."**
> **Response:**
> We have corrected this inconsistency.
>
> ---
>
> ### **Reviewer: "Please improve Algorithm 1 to clarify the following: (a) Is the input a whole utterance or a T-F bin/block? (b) Does the embedding represent a single patch or a sequence of embeddings for an utterance? (c) Why is line 12 and 13 indexed incorrectly?"**
> **Response:**
> To improve clarity, we have made the following updates to **Algorithm 1**:
>
> 1. **Clarified Input Representation (a & b):**
>    - We explicitly stated in the algorithm comments that the input **\( x \)** corresponds to **T-F bin patches or time-domain blocks**, rather than a **whole utterance**.
>    - We also clarified that **\( h \) and \( z \)** are **embeddings for individual patches or blocks**, not embeddings for **entire utterances**.
>    - This ensures that the reader understands that our method **operates at the patch level** rather than on full sequences.
>
> 2. **Corrected Indexing Issue (c):**
>    - We have fixed the **incorrect indexing in line 12**, ensuring consistency with the **notation used throughout the algorithm**.
>
> ---
>
> ### **Reviewer: "Notation in Sec 2.2 is confusing. Are Equations 6 and 7 the same as Equations 9 and 10? Why does taking softmax create an assignment matrix in Equation 8?"**
> **Response:**
> We thank the reviewer for the insightful question. The transformation from **\( \bar{F} \in \mathbb{R}^{n \times d} \)** to **\( S \in \mathbb{R}^{n \times k} \)** follows a **linear mapping and softmax normalization** to obtain probabilistic cluster assignments.
>
> #### **Feature Representation \( \bar{F} \):**
> The matrix \( \bar{F} \in \mathbb{R}^{n \times d} \) represents the **feature embeddings** extracted from a pre-trained model, where:
> - \( n \) is the **number of patches** (T-F bins or speech blocks).
> - \( d \) is the **feature dimension**.
>
> #### **Linear Mapping to Cluster Space:**
> To transition from the **feature space** to the **clustering space**, we apply a **linear transformation** using a weight matrix **\( W \in \mathbb{R}^{d \times k} \)**:
>
> \[
> Z = \bar{F} W
> \]
>
> where:
> - \( Z \in \mathbb{R}^{n \times k} \) contains **raw cluster assignment scores** before normalization.
>
> Alternatively, in **spectral clustering**, \( Z \) is obtained by computing the **top \( k \) eigenvectors** of the **modularity matrix** \( B \), storing them as columns.
>
> #### **Softmax Normalization to Obtain \( S \):**
> The cluster assignment scores \( Z \) are converted into **probability distributions** using a **row-wise softmax**:

---

### Review · Reviewer_zXGy · 2024-12-06

**Summary Of Contributions:**

The authors formulate the problem of speech separation as an instance of graph clustering. In this formulation, the nodes in the graph represent signals that are localized in time and frequency, and the goal is to partition the nodes according to the dominating speaker in each signal.  To solve the problem in this way, it is necessary to define a graph and then to optimize an objective function for graph partitioning. The graph is obtained by extracting feature vectors from a contrastive embedding (the pre-trained model in section 2.1) and defining an adjacency matrix from thresholded dot products of these feature vectors. These feature vectors are also used as inputs to a recurrent neural network (the BLSTM model in section 2.2); this network outputs a partition assignment matrix, and it is trained to maximize a regularized "modularity" that measures the quality of the graph partitioning based on this assignment matrix.

The authors show that this approach performs very well on multiple data sets (described in section 4). As one might expect, the results are better when the approach uses time-domain blocks of speech signals instead of time-frequency patches; the latter suffer from to phase-reconstruction issues (section 5.2). The results are also better when the pre-trained embedding is further optimized and "fine-tuned" to optimize the modularity objective (section 5.1). The paper is self-contained and complete: in addition to evaluating their own approach, the authors summarize previous work in this area (section 1.1) and compare to existing baselines (section 5 and 6).

**Audience:**

Yes

**Broader Impact Concerns:**

None.

**Claims And Evidence:**

Yes

**Requested Changes:**

The paper is impressively clear, but the following are suggestions for improvement.

* Please increase the size of Figure 1, which is barely readable on the printed page.

* On page 5, can you explicitly define the similarity term, sim$(z_{n_i},z_{nr_i})$? Do you simply mean the inner product?

* On page 6, in eq. (5), why do you use the Frobenius norm rather than the squared Frobenius norm (which seems more typical)?

* On page 6, is it possible to avoid a notation clash between the partition assignment matrix ($S$) and the TF bin patches ($S_n$)?

* On pages 6-7: I was thrown off course by eqs (7-8). From what I can tell, these equations are included to describe previous work, only then to be superseded immediately by eqs. (9-10), which describe what the authors actually do. Consider omitting eqs. (7-8) or framing these equations more carefully in the surrounding text (e.g., using separate paragraphs to distinguish the past and current approaches).

* On page 7: after eq (10), the authors describe how 3x3 patches are used to obtain 1x1 TF bins. This description felt out of place; I thought it should come earlier.

* On page 7: perhaps spell out bidirectional long-short-term memory before using the acronym (BLSTM)?

* On pages 7-8: I would have preferred this discussion (section 2.3) of the adjacency matrix before the description of the objective function for deep modularization (2.2). It would have been helpful to define the graph clearly (i.e., its nodes as 1x1 TF elements and its adjacency matrix A) before defining the modularity in terms of these quantities. As currently written, sections 2.2-2.3 were not as clear as the rest of the paper on a first reading.

**Strengths And Weaknesses:**

The paper has many strengths. Though speech separation has been previously studied as a problem in graph partitioning, it seems novel how the authors construct the adjacency matrix of the graph (from a pre-trained contrastive embedding) and how the authors perform the partitioning (by optimizing the regularized modularity via deep learning). Also, the writing is very clear (except in a few places), and the results are competitive. The solution in this paper has many different parts -- the preprocessing of the speech signal into TF bins or time-domain blocks, the contrastive embedding of the pre-trained model, the graph partitioning by deep learning, the regularized modularity as an objective function, the graph construction, the details of learning, the manner of evaluation, etc. Many details are provided to make the work reproducible, and yet, despite this complexity, the paper reads quite well.

Arguably the weakest part of the paper is the conclusion. Perhaps the authors can critically examine the weakest links in their approach and offer directions for future research?

---

> ### Author Response · Authors · 2025-01-06
> **Response**
>
> Reviewer’s comment: Please increase the size of Figure 1, which is barely readable on the printed page.
>
> Response: We have improved Figure 1 by increasing its size and enhancing clarity for better readability in the printed version.\
>
> Reviewer’s comment: On page 5, can you explicitly define the similarity term, sim? Do you simply mean the inner product?
>
> Response: We have explicitly clarified on page 5 that the similarity term, sim, refers to the inner product.
>
> Reviewer’s comment: On page 6, in eq. (5), why do you use the Frobenius norm rather than the squared Frobenius norm (which seems more typical)?
>
> Response: Thank you for the insightful comment. We used the Frobenius norm in Eq. (5) because this formulation was directly adopted from the cited previous work. Our goal was to maintain consistency with that approach to ensure comparable results and avoid introducing changes that might alter the interpretation of the findings. While we acknowledge that the squared Frobenius norm is more typical in certain contexts, we chose to retain the original formulation to align with prior methodologies.
>
> Reviewer’s comment: On pages 6-7: I was thrown off course by eqs (7-8). From what I can tell, these equations are included to describe previous work, only then to be superseded immediately by eqs. (9-10), which describe what the authors actually do. Consider omitting eqs. (7-8) or framing these equations more carefully in the surrounding text (e.g., using separate paragraphs to distinguish the past and current approaches).
>
> Response: We have removed Equations (7) and (8) and rewritten the section on pages 6 and 7 to improve clarity and incorporate your suggestions.
>
> Reviewer’s comment: On page 7: after eq (10), the authors describe how 3x3 patches are used to obtain 1x1 TF bins. This description felt out of place; I thought it should come earlier.
>
> Response: Since we obtain 1×1 1×1 T-F bins after processing the  3×3 patches, we believe the current location of this discussion enhances the logical flow of the explanation. However, we are open to further suggestions if needed.
>
> Reviewer’s comment: On page 7: perhaps spell out bidirectional long-short-term memory before using the acronym (BLSTM)?
> Response: We have corrected this as advised by spelling out Bidirectional Long Short-Term Memory (BLSTM) before using the acronym.

---

### Review · Reviewer_yXRG · 2024-12-17

**Summary Of Contributions:**

The paper introduces a speech separation pipeline integrating a pre-trained model and deep modularization for clustering speech signals. Empirical results on datasets (WSJ0-2mix, WHAM!) demonstrate the technique’s competitiveness, scalability, and generalization to noisy and reverberant environments.

**Audience:**

Yes

**Broader Impact Concerns:**

I did not observe ethical concerns.

**Claims And Evidence:**

Yes

**Requested Changes:**

1. Can you evaluate the impact of using sub-optimal pre-trained encoders? How does the separation performance degrade if the pre-training corpus is smaller or noisier?
1. Visualizing cluster assignments (e.g., embedding spaces) may help understand how the model separates speakers intuitively.

**Strengths And Weaknesses:**

Strengths:
1. The method elegantly avoids the computationally expensive PIT-based models, replacing them with clustering-based approaches. It also avoids parallel annotated datasets.
1. The use of a pre-training model with contrastive learning is novel.

Weaknesses:
1. The reliance on pre-trained models introduces a potential limitation. Performance might degrade without a well-trained encoder. This aspect is only lightly discussed.
1. While raw speech blocks perform better, the reasons for the T-F bin underperformance could be explored in more detail.
1. Despite MOD4’s strong performance, it is unclear whether the improvements are due primarily to the pre-trained encoder or the deep modularization component.

---

> ### Author Response · Authors · 2025-01-06
> **Response**
>
> Reviewers Comment: The reliance on pre-trained models introduces a potential limitation. Performance might degrade without a well-trained encoder. This aspect is only lightly discussed.
> Reviewer's comment: Despite MOD4’s strong performance, it is unclear whether the improvements are due primarily to the pre-trained encoder or the deep modularization component.
>
> Response: Thank you for your valuable comment. We acknowledge that the reliance on pre-trained models introduces a potential limitation, as performance may degrade when using a poorly trained or noisy encoder. While we lightly discussed this aspect in the original manuscript, we have now expanded the discussion in the revised version to better emphasize the importance of encoder quality.
>
> Additionally, we believe that our ablation study, which compared using a frozen encoder versus fine-tuning the encoder during clustering, demonstrated the critical role of encoder quality in achieving effective separation. Specifically, we trained our modularization module (MOD) under both configurations, allowing us to assess whether fine-tuning would improve clustering performance or if a frozen encoder could generalize effectively without additional updates. The results indicated that while fine-tuning improved performance in some cases, the frozen encoder also exhibited strong generalization, suggesting that the approach is relatively robust to varying encoder quality.
>
> Furthermore, we have proposed future work that could explore methods for improving robustness to encoder variability, such as fine-tuning pre-trained models on domain-specific data or incorporating encoder-agnostic training strategies
> Reviewers comment: While raw speech blocks perform better, the reasons for the T-F bin underperformance could be explored in more detail.
> Response: In pg 12 we expand reasons on why T-F bins produces inferior results as compared to raw blocks.
>
> response: Thank you for your thoughtful comment. We acknowledge the importance of determining whether the performance improvements in MOD4 are primarily driven by the pre-trained encoder or the deep modularization component (MOD). While our current work focuses on the overall performance of the combined system, we agree that further isolating the contributions of the encoder and the modularization component would provide valuable insights.
>
> To partially address this concern, we conducted experiments using both a frozen encoder and a fine-tuned encoder during clustering, as described in Section 5.2. This allowed us to assess the impact of encoder fine-tuning on performance. We observed that fine-tuning generally improved separation quality, indicating that the encoder plays a significant role. Specifically, the fine-tuned encoder with raw speech blocks (MOD4) outperformed the frozen encoder with raw speech blocks (MOD2), reinforcing the importance of the encoder in achieving high-quality separation.
>
> To quantify the contribution of deep modularization, we compared MOD with a baseline clustering method (K-means) using the same encoders. These experiments demonstrated that deep modularization contributes positively to performance, beyond the capabilities of the encoder alone.
>
> In future work, we plan to explore additional ablations that more directly isolate the impact of each component, such as testing different modularization architectures or using alternative pre-trained encoders

---

### Decision · Action_Editor_9eiP · 2025-02-02

**Recommendation:** Reject

**Comment:**

Based on the reviewers feedback and discussion, missing responses to critical questions from reviewer vHhJ and unclear behaviour between T-F bins and raw chunks, I recommend rejection of the paper.

I suggest authors to revise the paper, and one could consider the following way of revision:
- Rework the paper that proposed method is supervised one
- Make main contribution as modularization with showing that it is better than Deep Cluster approach
- Make raw bins central in the paper, showing that all prior works could benefit from switching from T-F representations to raw wave representations.

Regarding if the method is supervised or not -- I could agree that the algorithm itself may not use speaker labels and exact mixture, however this doesn't change the fact that training data should have 1 speaker clean speech only. Authors can say that the algorithm itself may not use particular properties as others do, but this doesn't mean that comparison with supervised methods should be then neglected -- as the data will be of the same type, fair comparison is comparison with supervised methods and then the claim should be that you can outperform them. Otherwise, I don't see the point why I should be using your methods instead of prior works.

**Audience:**

The paper will be relevant to machine learning and speech community given the new variant for speech separation as well as some ablations (like modularization vs knn, raw speech feature extraction or T-F bins).

**Claims And Evidence:**

The paper proposes an unsupervised speech separation method with improved time complexity avoiding all permutations of the sources via deep modularization. While reviewers pointed out novelty of the SSL usage for speech separation, reduced complexity, stronger performance than KNN and close performance to supervised baselines, there are several critical claims which need major paper revision even after discussion (e.g. authors did not respond to the critical questions and requested changes from reviewer vHhJ):
- (major) "Despite MOD4’s strong performance, it is unclear whether the improvements are due primarily to the pre-trained encoder or the deep modularization component." -  reviewer yXRG. While authors did ablation on KNN showing that modularization has the effect, the main improvement is still coming from switching from T-F bins to raw waves. Besides as pointed out by reviewer vHhJ, there is no proper comparison with Deep Cluster methods (e.g. changing modularization to Deep Clustering objective to show that modularization is the source of the improvements). Right now, results show that the proposed method can outperform others only if raw speech blocks are used, which is not a central focus in the paper and not a fair comparison with the prior works then.
- (major) There are no controlled experiments with MixIT both on quality and training time (reviewer vHhJ). As authors motivated in the introduction, complexity of MixIT will be reduced by proposed method, but there is no any discussion on that and authors did not comment on that request.
- (major) While authors claim to be unsupervised, it is not true, as they use the clean speech (1 speaker speech) for pretraining the encoder, which is vital for the method still: even if mixture and speaker labels are not used in the method itself (this is what authors consider unsupervised). Clarification on that was asked by reviewer vHhJ. Having clean speech with 1 speaker only is very strong supervision and allows to perform all variants of mixtures to train all prior supervised methods. There will be also assumption that we have a lot of such data available for different conditions to be able to train robust encoders. Thus, we will be able to train wavesplit or other supervised-based prior works on the same amount and diversity of data. Thus, comparison with all supervised baselines is unfair (as current paper uses some data for SSL pre-training), moreover we expect that proposed method should compare only with supervised methods and should outperform them which is not the case. **The same main issue with the paper was summarized in previous submission to TMLR by another Action Editor,** which is still unresolved.
- (minor) Significant performance difference between raw speech and T-F representation pointed out by reviewer yXRG. While authors added some textual clarifications, STFT is a very robust representation of speech established over decades. Even in the SSL domain in speech it was shown that MFCC, MFSC are on par with raw wave for SSL training, thus the observed gap needs more investigation.

**Resubmission Of Major Revision:**

The authors may consider submitting a major revision at a later time.